# Dendritic Cells and CCR7 Expression: An Important Factor for Autoimmune Diseases, Chronic Inflammation, and Cancer

**DOI:** 10.3390/ijms22158340

**Published:** 2021-08-03

**Authors:** Emma Probst Brandum, Astrid Sissel Jørgensen, Mette Marie Rosenkilde, Gertrud Malene Hjortø

**Affiliations:** Department of Biomedical Sciences, Faculty of Health and Medical Sciences, Blegdamsvej 3B, Room 18.5.32., DK-2200 Copenhagen, Denmark; emma.brandum@sund.ku.dk (E.P.B.); asj@sund.ku.dk (A.S.J.); rosenkilde@sund.ku.dk (M.M.R.)

**Keywords:** dendritic cell, CCR7, chronic inflammation, MS, RA, psoriasis, cancer

## Abstract

Chemotactic cytokines—chemokines—control immune cell migration in the process of initiation and resolution of inflammatory conditions as part of the body’s defense system. Many chemokines also participate in pathological processes leading up to and exacerbating the inflammatory state characterizing chronic inflammatory diseases. In this review, we discuss the role of dendritic cells (DCs) and the central chemokine receptor CCR7 in the initiation and sustainment of selected chronic inflammatory diseases: multiple sclerosis (MS), rheumatoid arthritis (RA), and psoriasis. We revisit the binary role that CCR7 plays in combatting and progressing cancer, and we discuss how CCR7 and DCs can be harnessed for the treatment of cancer. To provide the necessary background, we review the differential roles of the natural ligands of CCR7, CCL19, and CCL21 and how they direct the mobilization of activated DCs to lymphoid organs and control the formation of associated lymphoid tissues (ALTs). We provide an overview of DC subsets and, briefly, elaborate on the different T-cell effector types generated upon DC–T cell priming. In the conclusion, we promote CCR7 as a possible target of future drugs with an antagonistic effect to reduce inflammation in chronic inflammatory diseases and an agonistic effect for boosting the reactivation of the immune system against cancer in cell-based and/or immune checkpoint inhibitor (ICI)-based anti-cancer therapy.

## 1. Introduction

The chemokine system encompasses ~40 chemokines signaling through 18 chemokine receptors. It is characterized by a high degree of promiscuity with one ligand signaling through multiple receptors and with one receptor binding to and becoming activated by multiple chemokine ligands [1].

As the name infers, chemokines play important roles in controlling immune cell migration and positioning [2]. The unique ability of CCR7 to coordinate the meeting between activated dendritic cells (DCs) and various T cell subsets, including naïve, regulatory, and memory T cells, places CCR7 and its ligands in control of a central immune hub effectively controlling the onset of a diverse set of immune responses depending on the conditions, including inflammation, tolerance, memory, and autoimmunity [3]. CCR7 is involved in the progression of multiple diseases and could be a potential future drug target for halting disease progression in especially chronic inflammatory diseases [4,5].

In this review, we initially focus on the CCR7-CCL19/CCL21 axis and how it controls DC mobilization and T-cell activation. We then dive into its role in the establishment and progression of selected autoimmune diseases, including multiple sclerosis (MS), rheumatoid arthritis (RA), and psoriasis. Lastly, we revisit the binary role CCR7 plays in combatting and progressing cancer, and we discuss how CCR7 and DCs can be harnessed for the treatment of cancer.

## 2. Differential Roles of CCR7 Ligands in DC Mobilization

Dendritic cells (DCs) are highly versatile antigen-presenting cells (APCs) capable of capturing and processing antigens (Ags) to initiate adaptive immune responses through the action of Ag presentation and T-cell priming in the lymph node (LN) [6].

The mobilization of DCs to lymphoid organs has previously been recognized to be dependent on chemokine stimulation and the activation of chemokine receptor CCR7 [7,8], a G-protein-coupled receptor expressed on various immune cells, such as DCs as well as naïve, regulatory, and memory T cells [3]. CCR7 is regulated by its two endogenous ligands; the CC-chemokines CCL19 and CCL21, and a third naturally C-terminal truncated version of CCL21, i.e., Tailless-CCL21 [9]. Both CCL19 and CCL21 are expressed by stromal cells of the LN and by high endothelial venule (HEV) cells [10,11]. 

Moreover, CCL21 is secreted by afferent lymphatic vessels, whereas CCL19 is secreted by activated DCs in the LN [10,12]. CCL19 and CCL21 only share an amino acid identity of 32%, and CCL21 greatly differs from CCL19 due to its positively charged 32-residue extended C-terminal tail [13,14]. The elongated tail enforces a less active conformation of CCL21, i.e., an auto-inhibited version [15]. The third ligand, Tailless-CCL21 is generated from CCL21 upon cleavage by DC-released proteases, leading to a conformation change in CCL21 that improves its potency [9].

Although these three naturally occurring ligands share the same cognate receptor, they interact differently with CCR7, resulting in distinctive cellular responses [1,16,17] (Figure 1). CCL19 induces a strong and short-lived signal, whereas CCL21 induces a weak but persistent signal. Thus, CCL19 induces efficient signaling of CCR7 through G_αi_ and allows for subsequent β-arrestin recruitment and CCR7 internalization. In contrast, CCL21 is a weak agonist of both pathways and only induces a low level of receptor internalization [1,16,18]. CCL19 and CCL21 elicit different DC chemotaxis responses, as CCL19 was revealed to be 10–100-fold more potent than CCL21 in inducing the directed migration of these cells [19,20].

Based on their differential expression pattern and activity in diverse signaling pathways, it is fair to assume that the natural ligands of CCR7 play different roles in both homeostasis and immunity, and that the in vivo cleavage of CCL21 by DC-released proteases may turn CCL21 into a more robust signal that, although, due to its internalization via atypical chemokine receptor 4 (ACKR4), is also short-lived like CCL19 [21,22]. 

Given its expression by lymphatic endothelium, CCL21 is considered the major LN-homing chemokine for tissue resident DCs [23], whereas both CCL19 and CCL21 participate in the homing of T cells and plasmacytoid DCs through HEVs [10,11]. Due to the short-lived nature of CCL19-induced signaling, this chemokine is believed to play a role in the scanning process, coordinating multiple DC–T cell encounters during T cell priming in the LN [24] (Figure 2).

## 3. DC Maturation and Tissue Egress

DCs may reside within tissues (tissue resident DCs) or be recruited to the site of inflammation as a result of inflammatory cytokines [25]. At the site of inflammation, DCs encounter pathogen-associated-molecular patterns (PAMPs) (exogenous ligands), or damage-associated molecular patterns (DAMPs) (endogenous ligands). PAMPs and DAMPs are recognized by pattern recognition receptors (PRRs) expressed on DCs, such as cell surface C-type lectin receptors (CLRs) [26], toll-like receptors (TLRs), and intracytoplasmic NOD-like receptors (NLRs) [27,28].

Ag uptake, through phagocytosis and micropinocytosis, together with PRR stimulation leads to DC maturation; converting immature DCs (iDCs) to mature DCs. The maturation of DCs involves several cellular changes, which reflect the DCs transition from an Ag-sampling cell to an Ag-presenting cell where the DCs present the sampled Ags on major histocompatibility complex I or II (MHCI/II) molecules to T cells [29,30]. DC maturation involves changes within the MHC and Ag processing, the migratory abilities, as well as the metabolic processes to accommodate these functions.

MHCI molecules are expressed on all nucleated cells and are primarily involved in the presentation of intracellular Ags including those from intracellular pathogens (e.g., viral) and normal self-antigens (self-Ags) expressed by the cells. Whereas MHCI molecules are constitutively expressed by most cells, MHCII expression occurs as part of a central gene alteration that is induced upon Ag-induced maturation of APCs [31].

MHCII is only expressed on APCs, e.g., dendritic cells, and is involved in the presentation of extracellular Ags (e.g., bacteria), cancerous Ags, and normal self-Ags.

iDCs derived from lymphoid organs constitutively presents self Ags on MHCII molecules to allow antigenic self-memory. However, these self-Ag-presenting MHCII molecules are only expressed transiently on the cell surface, then subsequently degraded, and the iDCs stay immature [32]. When iDCs encounter a pathogen and are exposed to, e.g., a PAMP, the synthesis of MHCII is downregulated, MHCII molecules in the endocytic compartment pool are recruited and expressed on the surface, and then DC maturation is initiated [31,32]. 

Depending on whether DCs encounter self or non-self Ags, they may promote a regulatory or an inflammatory response in T cells. DC discrimination between Ags relies on associated innate immune-suppressive or activating signals [33]. Cross-presentation allows the presentation of Ags normally presented on MHCII to be presented on MHCI molecules, thus, initiating a cytotoxic CD8^+^ T cell response against these Ags [34].

The stimulation of TLR with inflammatory signals promotes a transition in the cellular metabolism of DCs; a switch from mitochondrial respiration to glycolysis [35,36]. This shift to the glycolytic metabolism is found to be essential for CCR7 oligomerization, which is essential for DC migration [37] and de novo fatty acid synthesis, which is required to enable the expansion of the endoplasmic reticulum (ER) and Golgi, to support the production and secretion of proteins central for the activation of DCs [35,37,38,39]. This is required for DCs to preserve their characteristic dendritic cell morphology and migrate efficiently.

A shift in the chemokine receptor and integrin profiles accommodates the chemotaxis of activated mature DCs to the LN and supports the DC tissue egress. For instance, the chemokine receptor CCR6, which facilitates the recruitment of iDCs to the site of inflammation, is downregulated, whereas the CCR7 and CXCR4 expression is upregulated [40]. This shift in the chemokine receptor expression profile allows the DCs to leave the inflamed tissue and migrate towards the major LN homing chemokine CCL21 that drives the co-localization of mature DCs and naïve T cells in these organs, to accommodate T-cell activation [3]. 

Another important shift in the surface molecule expression profile going from immature to mature DC occurs within the integrin family; therefore, DC maturation is linked to a downregulation of the α5β1 integrin, making them less adhesive and thus promoting a migratory phenotype [41]. In fact, DCs can migrate in the absence of integrins through flowing and squeezing [42].

Once matured, CCR7-expressing DCs follow haptotactic chemokine gradients of CCL21 toward the lymphatic vessels and LN [43]. The ability of activated mature DCs to relocate to the LN is of extreme importance. Previous studies have revealed that DC subsets migrate into draining lymph nodes from the periphery via two pathways: (1) tissue-resident cDCs migrate into LNs through afferent lymphatic vessels and (2) pDCs circulating in the blood enter through high endothelial venules [44]. In both instances, CCR7 and its ligands are important for the LN entry.

Factors shaping the chemokine receptor and ligand activity play important roles in controlling the LN homing of DCs and, thus, in the mounting of immune responses. Inside the lymphatic vessels, CCL21, expressed by the lymphatic endothelium, may be cleaved by proteases released by the migrating DCs to produce soluble CCL21 (Tailless-CCL21), which is more potent than the full-length version of the chemokine [15,19,45,46]. The atypical chemokine receptor ACKR4 (also known as CCRL1, CCR11, or CCX-CKR) regulates DC migration towards and positioning in the LN through the scavenging of CCL21 and CCL19 and, thereby, contributes to maintaining functional chemokine gradients through the internalization and lysosomal degradation of the chemokines [21,22].

## 4. CCR7 and Induction of Organ Associated Lymphoid Tissues (ALT)

CCR7 and its ligands are important for the neogenesis and formation of lymphoid organs, such as lymph nodes. The normal segregation of B and T cells in their respective compartments within lymphoid organs depends on chemokines, with CCL21 (and to some extent CCL19) directing cell migration to the T cell zone through CCR7 and CXCL13 directing cell migration to the B cell zone through CXCR5 [10,47,48,49,50].

In addition to primary lymphoid organs, e.g., the thymus and bone marrow, and secondary lymphoid organs, e.g., the lymph nodes and spleen, a third type of lymphoid organs exists. These structures are termed tertiary lymphoid organs (TLO) or inducible lymphoid organs and are the accumulations of lymphoid cells within other organs. TLO structures emerge in infection and cancer but also in chronically inflamed tissue or at sites of autoimmunity and have been identified in a variety of autoimmune diseases. Tertiary lymphoid organs do not have a fixed structure and location, nor do they possess well-defined capsules as with the primary and secondary lymphoid organs.

However, TLO appear to mimic the function of the lymph nodes [51]. Inflammatory cytokines, such as members of the TNF family, can lead to the ectopic expression of the chemokines CCL21, CCL19, and CXCL13 [52,53]. The ectopic expression of CCL21 or CXCL13 can stimulate the formation of TLO with distinct B- and T-cell zones [53,54,55]. Other classical lymph node structures are present in the TLOs [56,57], such as the HEV from where blood-derived lymphocytes enter.

The tertiary lymphoid structures can partake in sustaining chronic inflammatory conditions, such as rheumatoid arthritis, Hashimoto thyroiditis, and Sjøgren’s syndrome [58,59,60,61], but may also be important for organ immunological homeostasis, which is the case for MALT (mucosa-associated lymphoid tissue). MALT structures may be preprogrammed but can expand upon inflammation and, thus, support immune responses when required. They disappear when the inflammation is cleared [62]. 

BALT, bronchus-associated lymphoid tissue, is a type of MALT formed within the lungs with organized lymphoid structures with HEVs and B- and T-cell areas, which occurs as a response to pulmonary inflammation [63]. The formation of BALT is necessary for efficient vaccination towards airborne Ags and respiratory immunological homeostasis, and DCs play an important role in the maintenance of the structures [63,64]. 

The skin is constantly exposed to various Ags, and TLOs are important during the immune surveillance of the skin where inducible skin-associated lymphoid tissue (iSALT) plays a central role in the induction of cutaneous adaptive immunity [65]. In addition, iSALT participates in the pathology of psoriasis [65], where CCR7 and CCL19 play important roles in the sustainment of these structures (see the section on psoriasis).

## 5. DC Subsets and Their Immunological Roles

Various types of DCs exist, which all, to various extents, may participate in the pathology of inflammatory diseases. Constant scientific and methodic advances increase our ability to distinguish between different DC subtypes and continuously expand the DC atlas. Morphology, surface markers, location, and function have all been used to characterize DC types. Later, developmental origin has been used to understand the relationship between DC subtypes, and recently experimental advances, such as single cell RNA sequencing, have been used in an attempt to classify the diversity of DC populations in an unbiased manner and have revealed additional subtypes [66,67,68].

The lack of uniquely expressed surface markers and, instead, the expression of shared surface molecules has made distinguishing between some DC subtypes, such as cDC2 and moDC, difficult and challenges the full understanding of what role individual DCs subpopulations play. In the following section, we provide a brief overview of some of the most important and defined DC subsets (Figure 3).

Different models of hematopoiesis and DC development have been proposed. In both mice and humans, DCs originate from hematopoietic progenitors, giving rise to multipotent progenitors (MPPs), subsequently generating different DC subtypes [70]. DC subtypes include conventional DCs (cDCs), further dived into cDC1s and cDC2s, also referred to as classical or myeloid, DCs (cDCs/mDCs); monocyte-derived DCs (MoDCs); and plasmacytoid DCs (pDCs) [71]. All DC subtypes display unique features. The differentiation of these is controlled by various transcription factors.

The conventional DCs, cDCs are subdivided into two major subsets determined by their ontogeny: type 1 cDCs (cDC1s) and type 2 cDCs (cDC2s), also referred to as CD141^+^ DCs, and CD1^+^ DCs, respectively. cDCs are derived from a common DC progenitor (CDP), that differentiates into pre-DCs [72,73,74]. Pre-DCs leave the bone marrow, transiently circulate in the bloodstream and inhabit lymphoid and non-lymphoid tissues, where they further differentiate into cDC1s and cDC2s [75,76,77,78].

Human cDC1s (CD141^+^) are found in the blood, tonsils, bone marrow, spleen, lymph nodes, and non-lymphoid tissues, such as the intestines, skin, liver, and lungs. cDC1s present exogenous Ags and secretes IL-12, leading to a cell-based immune response. They present Ags on MHCII to stimulate the activation of CD4^+^ T helper type 1 cells (Th1), and they are specialized in the cross-presentation of exogenous Ags on MHCI to stimulate the activation CD8^+^ T cells (cytotoxic killer cells) [79,80].

Human cDC2s (CD1c^+^) are found in tissues, lymphoid organs, and blood and are associated with CD4^+^ Th cell responses. Depending on the inflammatory milieu, cDC2 can skew the immune response towards various CD4^+^ helper cell types involved in different immune responses through Ag presentation on MHCII, including Th1 (only in human, mice elicit Th1 by cDC1s), Th17 (autoimmunity), Th2 (humoral), Tfh (T cell help to B cells), and Treg (tolerance) [69,81]. 

The varying T cell skewing may be a result of additional cDC2 subsets, such as the newly identified cDC2A and cDC2B defined by mutually excluding the expression of RORγt and T-bet, which display anti-inflammatory and pro-inflammatory traits, respectively [66,68]. The cDC original markers CD141 (cDC1) and CD1c (cDC2) have limitations as both are induced on cDC and monocyte-derived cells in tissues and in culture. Thus, distinguishing cDC2s in particular from moDCs can be difficult, as these subpopulations share most of the same surface markers, e.g., CD11b during inflammation [75,82].

MoDCs or TNFα- and iNOS-producing DCs (TipDCs), are “inflammatory dendritic cells” induced from monocytes by infection and inflammation [75,83]. In humans, three different types of monocytes exist: the classical, the intermediate, and the non-classical. These are usually characterized according to the surface expression of CD14 and CD16: CD14^++^ and CD16^−^ (classical), CD14^++^ and CD16^+^ (intermediate), and CD14^+^ and CD16^++^ (non-classical) [84]. In mice, two different subtypes exist. These are discriminated by the expression of lymphocyte Ag 6C (Ly6C). 

A high expression of Ly6C (Ly6C^hi^) is associated with pro-inflammatory, as well as antimicrobial functions, whereas a low expression (Ly6C^low^) is associated with patrolling functions [76,85]. Ly6C^hi^ monocytes express high levels of CCR2 and low levels of CX_3_CR1 (CCR2^hi^ CX_3_CR1^low^). Murine Ly6C^hi^ monocytes are analogous to classical monocytes in humans, while murine CCR2^low^ CX_3_CR1^hi^ and Ly6C^low^ are similar to human non-classical monocytes. It has been proposed that the intermediate monocyte subtype is in transition from the classical subtype to the non-classical [85]. 

The exact subtype from which monocyte-derived DCs (MoDCs) arise in vivo is still weakly defined [86]. Nonetheless, they have been suggested to arise from Ly6C^hi^ monocytes [76] corresponding to the human classical monocytes. During inflammation, monocytes can differentiate into monocyte-derived inflammatory DCs (Inf-moDCs) at the site of inflammation following exposure to granulocyte-macrophage colony stimulation factor (GM-CSF) and IL-4 [87,88].

MoDCs enter tissues from the bloodstream upon binding to adhesion molecules expressed by endothelial cells upon inflammation to combat the invading pathogens. According to predominant models, moDCs mainly function at the site of inflammation rather than migrating toward lymph nodes. Yet, ex vivo studies of moDC have reported them to express CCR7; secrete IL-1, IL-12, IL-23, and TNF-*α*; and stimulate both CD4^+^ and CD8^+^ T cells [75]. It should be noted that the distinction between cDCs and moDCs has not always been accounted for, and thus sometimes the literature may not distinguish between these subpopulations, making the characterization of subtypes a bit obscure. Therefore, whether the DCs found important in various lesions are cDC or moDC may be unanswered, possibly rendering an undervalued important role for moDC [75].

Plasmacytoid DCs (pDCs) resemble plasma cells in their morphology due to an eccentrically positioned nucleus and a rich endoplasmic reticulum and Golgi apparatus. pDCs are specialized to secrete interferons as part of inducing viral immunity. Thus, pDCs patrol the bloodstream and peripheral lymphoid organs to combat viral and bacterial infections, and they display a unique capacity to produce type I interferon (IFN) when exposed to bacterial or viral components [89]. Like the previously mentioned DC subtypes, pDCs require a specific set of differentiation factors and express distinctive surface markers separate from cDCs, which are important for the production of type I interferon [82].

An extensive RNA sequence analysis of DC subpopulations by Villani et al. revealed a new DC subtype, which could be captured within both the traditionally pDC and CD1c gates [66]. This is in line with a similar analysis by See et al. the same year, who identified a pre-DC population sharing classical pDC markers [67]. This pre-DC population gave rise to both cDC1 and cDC2. Collectively, the two studies identified the same subtype, which was given the name Axl^+^ (ASDC) [66,67,90]. The surface marker CD303, which was originally considered a pDC marker, is one of the shared markers for pDC and pre-DC populations [67]. 

Similar mouse studies revealed a corresponding mouse DC population, although lacking the Axl, leading to the naming of a new transitional DC population (tDC) [91]. These studies imply that the traditional ways of isolating pDC would include contamination by the cDC populations, and that some of the functions previously ascribed pDC may be a result of the heterogeneous population. For example, the IL-12 production and ability to induce naïve T cell proliferation was abolished when researchers stimulated a pure pDC culture [67].

## 6. Dendritic Cell Induced T Cell Differentiation

A pro-inflammatory phenotype of mature DCs is often characterized by a high expression of MHCII and costimulatory molecules, such as CD40, CD80, CD83, and CD86 [32,92,93], although differences between DC subsets exists–pDC versus cDC (cDC1/2) [75]. These molecules are important for Ag presentation and T-cell activation in the LN. Various T cell subsets are primed and activated by DCs during Ag presentation. The DC-induced differentiation of T helper (Th) cells into distinct Th subsets is determined by the DC priming conditions, which are defined by the cause of the DC activation.

The priming process include Ag engagement with T cell receptors (TCR) on naïve CD4^+^ Th cells, as well as the specific expression of cytokines and co-stimulatory molecules [94]. Naïve CD4^+^ Th cells can differentiate into serval Th subtypes, including Th1, Th2, Th17, regulatory T cells (Tregs), and follicular Th cells (Tfhs), which are all associated with the expression of both a master transcription factor, a specific signal transducer, and activator of transcription (STAT) protein [95]. More recently, additional subtypes have been recognized; Th9 and Th22, which were named according to their expression of IL-9 and IL-22, respectively [96,97,98].

## 7. DCs in Pathology of Multiple Sclerosis

Multiple sclerosis (MS) is an autoimmune disease where the immune system attacks the myelin sheaths around the axons of the CNS, which ultimately causes neurological damage. Immune cells generally enter the brain via the choroid plexus (CP) into the cerebrospinal fluid (CSF) in the ventricular space or across the blood-brain barrier (BBB) into the postcapillary meningeal veins to the underlying brain parenchyma [99]. The role of CCR7 in the development of MS is still unclear; however, several studies suggested that CCR7 is a contributing factor during recurrent and progressive MS.

In healthy individuals, CCL19 is secreted by epithelial cells in CP and then accumulates in the CSF [100]. Most lymphocytes in the CSF in healthy individuals are CCR7-positive central memory T cells [101,102], suggesting that the CCL19-CCR7 chemokine receptor axis plays a role during normal immune monitoring of the brain (Figure 4). In patients with recurrent and progressive MS, elevated CCL19 levels have been found in CSF [100,103]. In experimental autoimmune encephalomyelitis (EAE) (a murine MS model), it has been shown that T cells access the CNS via extravasation from post-capillary venules [104] and that blocking CCR7 signaling reduces the binding of T cells to inflammatory venules of EAE brain sections [105]. 

CCR7-positive cells accumulate in perivascular cuffs and meningeal infiltrates of the brain in the EAE model [106]. These data suggest that CCR7 participates in memory T cell CNS infiltration during MS. Elevated levels of CCL19 in CSF in MS patients have also been shown to correlate with elevated levels of intrathecal IgG production [100], suggesting that CCR7 also plays a role in the differentiation and expansion of IgG-producing B cells in MS (Figure 4). CSF in MS patients is enriched in CCR7 positive T cells and DCs [102]; however, since T cells in MS lesions do not express CCR7, this indicates downregulation of CCR7 after BBB transmigration [107]. 

DCs in MS are pro-inflammatory and CCR7 positive, and their frequency correlates with MS-associated genetic risk factors, making them likely to contribute to pathogenic responses during MS [108]. CCR7 signaling stimulates IL-12 and IL-23 production in DCs, leading to stimulation of a Th1 and Th17 response, respectively, during subsequent T-cell activation [109]. A deficiency of CCR7 ligands was shown to protect against the development of EAE, mainly due to a reduced IL-23 induction of a Th17 T-cell response [109]. Based on the above data, it seems likely that blocking CCR7 signaling could reduce both a humoral and an immune cell-mediated pathogenic course in MS, making CCR7 a potential target for interference with MS.

## 8. DCs in Initiation and Sustainment of Rheumatoid Arthritis

Rheumatoid arthritis (RA) is a systemic autoimmune disease associated with articular inflammation and synovial joint damage with an increasing disability over time. RA leads to cartilage and bone destruction, and can, over time, also lead to the inflammation of other organs, thereby, causing systemic cardiovascular and pulmonary complications [110,111].

The inflammation of the joints is a result of leukocyte infiltration with the release of pro-inflammatory cytokines from both innate and adaptive immune cells, and similar to other autoimmune diseases, Th17 cells play an important role in RA development [112]. The autoimmunity of RA has a genetic predisposition linked to especially HLA genes [113] and is associated with the presence of autoantibodies—both Rheumatoid Factor (RF) and antibodies against cyclic citrullinated peptides (CCP) [110,113].

Given the role of the HLA genes and, thus, Ag presentation in RA, it is not surprising that DCs play a role in the initiation and sustainment of this disease (Figure 5). Thus, a positive clinical outcome is associated with the blocking of CD80/CD86 T cell co-stimulation [114]. The RA synovium contains a large fraction of so-called semi-immature DCs, a state that can facilitate an increased sampling of the environment by the DCs followed by Ag presentation of the sampled arthritogenic Ags [115,116]. These differently matured synovial DCs induce higher levels of T-cell activation than blood-derived DCs [117]. 

Within the joints, immature DCs are detected in the lining of the synovium, whereas mature CCR7^+^ DCs are found in perivascular lymphoid aggregates [115] where they form clusters with T cells around blood vessels and sometimes in follicular structures and even germinal centers (GCs) [118,119]. DCs derived from RA synovium show an increased expression of CCL19 [120], and, in some cases, SNPs within the CCL21 gene are associated with RA susceptibility [111]. 

The expression of both chemokines is linked to the organization of lymphocytic aggregates, indicating a role for these in the organization of lymphoid infiltrates [54,115]. In RA samples, the expression of CCL21 and or CCL19 is correlated with the presence of lymphocytic aggregates; here, chemokines were not detected in samples without aggregates but could be detected in all samples with aggregates. Moreover, a higher number of CCL21^+^ and CCL19^+^ cells could be detected in samples also containing GCs [115]. 

This is interesting given that DCs, in addition to their role in T-cell Ag presentation, also support the activation of B cells through the expression of BAFF [121]. Similarly, the detection of CCR7^+^ cells also correlates with the presence of lymphocytic aggregates [115]. In a mouse model, CCR7 KO was associated with a failure to form these discrete T and B follicular structures [122].

None of the existing rodent RA models fully recapitulates human RA in particular due to the species difference between the human and rodent immune systems. To overcome this, the use of different transgenic mouse models is emerging [123]. Despite this lack, some insight into RA does arise from the existing models, which mirrors different aspects of the disease. Using the collagen-induced arthritis (CIA) mouse model, CCR7 was proven necessary for the development of CIA-inflammation. Anti-CII antibody levels were significantly reduced in a CCR7 KO mouse, and restoration of CCR7 in only DCs restored the inflamed condition [4]. 

Similarly, using the same model blocking MMP-9 (matrix metalloproteinase 9) prevented the DC migration to the lymph nodes and the subsequent T- and B-cell activation and proliferation. However, this blockage affected the early stages of CIA RA but not the later synovial inflammation and tissue destruction indicating a central role for CCR7 and DCs in the early phase of RA establishment in the CIA model [44]. In a similar manner, KO of CCR7 in a modified Ag-induced arthritis (AIA) model reduced early stages of disease development, while the chronic inflammation score was higher compared to in the WT [122]. 

Although the models display discrepancies to the role of CCR7 in the later chronic inflammation phase, they all highlight the central role of CCR7 and DCs in the early phase of disease establishment. In addition to CCR7, which is expressed on mature DCs, CCR6 expressed on immature DC are also associated with recruitment of DC to the synovial cavity in response to CCL20 expressed by synoviocytes [115].

Dendritic cells are recruited into the joints by locally produced cytokines, or differentiate within the joint from progenitors in response to locally derived cytokines [124]. RA conditioned media leads to the activation of monocyte-derived DCs with an increased level of CD83 and CCR7, and this activation is also associated with a metabolic shift in the DCs favoring a glycolytic state [38]. Upon inflammation, accumulation of, in particular, the cDC population was detected within the synovium [116]. 

This synovial cDC population was characterized as a myeloid DC population based on the CD141 expression and was transcriptionally and functionally distinct from their peripheral blood counterparts, thus, leading Canavan and colleagues to suggest the presence of a bona fide DC population [117]. However, whether this is truly a cDC population or instead moDCs might be the question since CD141, and CD1c, previously used to describe cDCs are also expressed by moDC [75]. 

Indeed, similar to other pathological lesions, the monocyte-derived DCs appear to form an inflammatory DC population in RA lesions, and in fact, characterization of the RA synovial fluid showed that monocyte-derived DCs from the inflamed synovium were distinct from tissue-resident cDCs and could secrete Th17 polarizing cytokines [125]. DC matured ex vivo from patient-derived monocytes even expressed higher levels of IL-6/IL-23 and had a skewed T cell polarization with a higher capability of inducing Th17 cells and a reduced induction of Treg compared to healthy controls [126].

In addition to the role of DC recruitment, studies suggest that CCR7 might also be involved in other RA pathogenesis processes, such as the association between CCR7 and increased M1 macrophage polarization with a correlated Th17 polarization and subsequent osteoclastogenesis [127]. CCR7 and CCL21/CCL19 may also play a role in neoangiogenesis and other microenvironmental changes [128,129], which supports the build-up of synovial inflammation [110].

## 9. DCs in Chronic Inflammation of Psoriasis

Psoriasis is an autoimmune disease within the skin, and possibly the most prevalent immune-mediated skin disease in adults [130]. The most common form, psoriasis vulgaris, is characterized by red scaly raised plaques, although different clinical forms exist. Histologically, the increased proliferation of keratinocytes leads to a thickening of the epidermis—the outer layer of the skin. Not only is psoriasis associated with this increased cell differentiation but also the differentiation program is altered resembling a state seen during wound repair. 

Here, aberrant terminal differentiation leads to plaque formation; a stratum corneum forms from incompletely differentiated keratinocytes, which do not lose their nuclei as in normal epidermis. The outer corneocytes fail to stack normally, adhere to one another, and secrete extracellular lipids creating scaling and breaks in the protective barrier [130]. Hyperplasticity and the dilation of blood vessels in the dermis causes redness (erythema) of the skin lesions. These changes are associated with immune cell infiltration into the skin [130] with neutrophils collecting in the epidermis and abundant mononuclear cells, myeloid DCs, and T cells collecting in the dermis [131].

Psoriasis can be triggered by many factors, including injury and trauma, infections, and medication [131]. Murine studies with the antiviral antitumor drug Imiquimod (IMQ) showed how medication can trigger the development of a psoriasis-like condition in mice through the activity of DCs. Here, IMQ leads to the activation of the toll-like receptors (TLR7/8), which are part of the innate immune system, and thereby initiate the inflammatory state [132]. 

Similarly, as a consequence of injury to the skin, cell death can trigger immune activation. Under normal conditions, genomic DNA is tolerated and would not lead to immune activation. However, keratinocytes in the skin produce several antimicrobial peptides, which are also part of the innate immune system. Studies from Michel Gilliet showed that at least one of these, the antimicrobial peptide LL37, can form complexes with DNA or RNA from dying cells. These complexes are then able to bind to a group of intracellular TLRs in plasmacytoid and myeloid DCs, leading to efficient activation of these cells [133,134] (Figure 6).

The cytokines IL-23/IL-17/IL-22 can play a role in disease development [55,132,135]. In psoriasis, IL-12 and IL-23 are mainly released by inflammatory myeloid DCs, leading to activation and expansion of Th17 cells [131]. Local IL-23 injection can, similar to the application of IMQ, induce a DC-dependent psoriasis-like state, and, in both human and mouse skin inflammation, this cytokine is highly expressed [55,132]. IL-23 promotes activation and expansion of Th17 cells [136], and the importance of effector IL-17 in psoriasis has been demonstrated in both humans and mice [137,138,139] indicating a central role for Th17 cells, although Th1 and Th22 cells may also participate in the inflammatory milieu [131]. 

Keratinocytes in the inflamed skin can participate in the initial recruitment of leukocytes [131,140] or maintain the chronic inflammation by responding to the leukocyte-expressed cytokines by upregulating different inflammatory mRNAs. These products in turn feedback to the immune cells to maintain chronic T-cell activation [131]. These inflammatory molecules originating from the keratinocytes are also thought to be important for the continued recruitment of leukocytes with relatively short life spans, such as neutrophils and DCs [131]. One of these is the chemokine CCL20 expressed by keratinocytes during epidermal injury [140] and its associated chemokine receptor CCR6, which was shown to mediate monocyte-trafficking into inflamed skin where they matured to monocyte-derived DCs [141] (Figure 6).

The identification of DC TLRs in the establishment of psoriasis has yielded much attention towards this group of proteins as therapeutic targets where several compounds have been investigated for their ability to ameliorate the TLR-induced inflammation, either through a direct effect on TLRs or through interference with downstream signaling mediators [142,143,144]. The antibiotic azithromycin can improve the severity of IMQ-induced mouse psoriasis by interfering with the lysosomal processing of TLR7 maturation and signaling in DC [145]. Blockage of JNK/c-Jun signaling, which is required for the TLR7-IMQ mediated inflammation but not for normal DC skin development, was found to reduce the DC IL-23 production and relieve the psoriasis-like skin inflammation [146].

Another aspect that may facilitate the existence of chronic inflammation is the presence of iSALT (inducible skin-associated lymphoid tissue) structures [65]. These dermal aggregates of immune cells are found in psoriatic lesions but not in normal skin, and primarily contain CD3^+^ T cells, CD11c^+^ LAMP3/DC-LAMP^+^ DC, and, to a minor extent, CXCR5^+^ B cells [147]. Both CCR7 and CCL19, but not CCL21, display upregulated gene expression in these lesions and are clearly identified in dermal aggregates [147]. 

The identification of co-expression between CCR7 and both CD3 and LAMP3 [147] and the fact that ectopic CCL19 expression can organize functional lymphoid structures [54] points to an important role for CCR7 and CCL19 in the establishment of these iSALT structures. Likewise, a potential role for CCR6/CCL20 in the recruitment of T cells and DC to these aggregates has also been suggested [148]. Although the IL-23 responding cells are T cells, DCs are critically required for the IL-23- (and IMQ-) induced skin lesions [141]. 

Studies of the different DC populations show that, although Langerhans cells are present in the epidermis and upon activation can migrate out of the epidermis and activate the T cells in the draining lymph nodes [131], resident Langerhans cells (LC) are expendable for disease development as with conventional dermal DCs [141,149]. LC in the epidermis can be reconstituted in two ways; the long-term where bone-marrow precursors reconstitute the LC compartment and the short-term where monocytes under inflammatory conditions can be recruited to the skin and proliferate and differentiate into LC [150,151]. 

In the IL-23 induced psoriasis model, this later group of monocyte-derived LC cells appears to be an important contributor to the inflammatory response, together with the monocyte-derived dermal DC population [141]. Similar monocyte-derived LC and dermal DC populations were identified in human psoriatic skin [141]. The critical DC population for psoriasis development, thus, seems to be TNF- and IL-1b-producing monocyte-derived LC and dermal DC entering the skin through CCR6-driven trafficking [141,148,149].

DCs in the iSALT structures secrete CCL19, which is important for sustaining these structures and attracting various CCR7^+^ T cell subsets. This suggests the potential situation where CCR7^+^ central memory T cells, which normally circulate between the blood and draining lymph nodes until they are primed by DCs within the lymph node [152,153], may instead undergo expansion to effector memory T cells within the skin iSALT structures by DCs. This scenario underscores the very important role of CCR7 and CCL19 in the disease progression of psoriasis.

## 10. The Binary Role of CCR7 in Combatting and Progressing Cancer 

CCR7 and its ligands play two important but contending roles in cancer. On one hand, the lymph node homing ability of CCR7 might promote LN metastasis of a variety of CCR7-expressing cancers, and a higher expression of the chemokines and receptor is associated with a poor prognosis. On the other hand, it has been hypothesized that CCR7 could be exploited as a means of potentiating immune cell trafficking to tumors.

It has been suggested that cancer metastasis is organ specific and that chemokine axes play important roles here with the CCR7-CCL19/CCL21 axis being particularly important for LN metastasis [154,155]. The overexpression of CCR7 appears to correlate both with LN metastasis in various cancers (breast, pancreatic, esophageal, lung, etc.), as well as a poorer survival prognosis [156,157,158,159,160,161,162,163].

However, discrepancies exist as CCR7 expression in lung cancers seems to correlate with a better survival prognosis [164]. Indeed, CCR7 is implicated in cancer; however, whether targeting CCR7 using agonists or antagonists for the intervention in cancer remains debatable. Some small molecule antagonists of CCR7 have been recognized thus far, including Navarixin and Cmp2105, which are now being tested in clinical trials for their metastasis-preventing activities [165,166].

Although it remains undetermined whether CCR7 agonists or antagonists should be used for anti-tumor activity, it is well established that the agonistic CCR7 activity of immune cells could yield a purposeful tool for increased antitumor immune activity. The intra-tumoral administration of CCL21 or CCL19 ligands induces increased intratumoral influx of DCs and T cells, ultimately, reducing tumor growth and prolonging the survival of tumor-bearing mice [167,168,169,170,171]. 

Although appearing a promising therapeutic approach, the clinical delivery of chemokines remains a challenge. The systemic delivery of CCL19 and CCL21 can lead to complications and toxicities, and thus a robust drug delivery system is required. One focus has been the use of hydrogel or nanoparticles to achieve tumor chemokine release [172,173,174]. Another approach is the introduction of transfected or transduced cells within the tumor that could locally produce and excrete the chemokine(s) [175,176]. 

A slightly different approach is to modify the immune cells themselves to overexpress CCR7 or CCL19/21. The expression of CCL19, together with IL-17, in CAR T cells showed promising results with an improved efficacy in a mouse model [177]. Improved anti-tumor properties have also been associated with the overexpression of CCR7 on various immune cells, e.g., NK cells [178] but also dendritic cells [179], with the later also showing greater migratory abilities towards the draining lymph node.

Dendritic cells themselves are also of great interest within the field of cancer immune therapy. During initial cancer development, cDC1s take up tumor-associated Ags (TAAs) and present them to naïve T cells and CD4^+^ T helper cells to induce a CD8^+^ cytotoxic T-cell response [180,181]. This principle has become key as a means to re-activate the immune system towards cancer. The ability to re-activate the immune system towards cancer has thus far been explored employing ample strategies. 

Current Immuno-oncology (IO) strategies involve immune checkpoint inhibitors (ICI) treatments with PD-L1-, PD-1-, and CTLA-4-blocking antibodies to hinder cancer cell immune evasion and favor T-cell activation [182]. The use of CAR T cells—patient T cells that are manipulated ex-vivo to generate chimeric Ag receptor-expressing T cells—is another approach to activate the immune system towards TAA [183]. Unfortunately, both strategies are currently only effective in a subset of cancer types with low overall effectiveness, and can eventually lead to relapse and renewed tumor growth [182,183,184,185,186]. 

Thus, different approaches, such as DCs, are being considered, and current clinical studies are exploiting the use of DCs as a mean to re-activate the immune system towards cancer by restoring failed of frustrated T-cell priming. Therefore, antibodies inducing DC maturation through CD40 activation as well as DC vaccines are considered potential future tools to initiate anti-tumor immunity alone or in combination with ICI [185,187,188] or CAR-T [189].

Currently, only a few DC vaccines are on the market; however, the field of cell-based immunotherapy has experienced a huge increase in interest, with many DC vaccination studies currently ongoing within multiple indications (clinicaltrials.gov). One problem with DC vaccines has been related to their limited clinical efficacy, which rarely exceeds 15% [190]. One major issue in finding efficient DC vaccines relates to the poor LN-homing ability of the ex vivo matured TAA presenting DCs upon administration. 

Many factors contribute to the low migratory capacity of DC used in vaccines, some of the factors to be aware of are the chosen maturation strategy, the amount of the injected DCs, and the injection site. Thus, earlier studies have revealed that some maturation strategies produce DC that are efficient in T-cell activation but display a poor migratory phenotype and vice versa [191,192]. In a study by de Vries et al., it was discovered that, upon the bolus injection of DCs, the injection of an overly large number of DCs had a negative effect on LN homing [193]. 

Researchers speculated that intranodal injection vs. subcutaneous injection of the DC vaccine could have a positive effect on DC vaccine efficacy; however, overall, these delivery routes seem evenly effective [194]. Thus, it appears that future optimization of maturation strategies together with advances in ways to boost DC LN homing may be critical for generating DC vaccines of high efficacy that can stand alone [195].

It is likely that tumor-specific T cells induced by adjuvant DC vaccination could result in a more potent tumor-specific immune response if the vaccination precedes ICI treatment in the metastatic setting. This was observed with the administration of ipilimumab (a monoclonal antibody targeting CTLA-4) in patients with relapse after adjuvant DC vaccination for stage III melanoma [196]. A similar effect was also observed retrospectively in patients with glioblastoma (GBM) that received chemotherapy after DC vaccination [197]. Thus, it seems that DC vaccination could be beneficial prior to ICI and CAR-T, for turning cold tumors hot and, thereby, susceptible to ICI treatment.

## 11. Conclusions

The CCR7-CCL19/CCL21 axis plays an important role in inflammation-driven disease progression. For this reason, drugs targeting CCR7 or its ligands could potentially be beneficial at certain time points during disease development to halt disease progression. Thus, it seems that antagonists of CCR7 would be beneficial for blocking immune cell recruitment to the inflamed tissue to prevent the exacerbation of autoimmune reactions in, e.g., the early phase of RA establishment. 

Such treatment would presumably counteract the accumulation of CCR7-positive DCs in the perivascular lymphoid organs in this disease as well as in the RA synovium. As the targeting of antibody-producing B-cells with Rituximab does not alter the disease progression [198], CCR7^+^ T cells together with CCR7^+^ DCs are likely involved in exacerbation inflammation through ongoing T cell priming driving the disease progression. 

An earlier study conducted in human tissue-SCID mice chimeras revealed that end-differentiated human memory T cells isolated from peripheral blood mononuclear cells (PBMCs) of RA patients, through maintained CCR7 expression, can home to lymphoid organs enhancing their survival, supporting clonal expansion, and perpetuating auto-reactivity [199]. Antagonism of CCR7 may also yield a purposeful tool in the prevention of lymph node metastasis of CCR7 overexpressing cancers [154,155]. Thus, CCR7 and its two ligands should be considered as possible targets for intervention with chronic inflammatory diseases, at an early stage. 

At the moment no drugs targeting CCR7 are on the market; however, recently, antagonists of CCR7 were disclosed in the literature. Both Navarixin and Cmp1220 inhibit signaling via CCR7 by stabilizing the receptor in an inactive state through interactions with the receptor from the intracellular side [166,200]. Whether the CCR7 antagonists can be used in disease control without adverse effects is unknown as CCR7 is important for upholding homeostatic immune functions [3]. Therefore, antagonists that selectively block part of the signaling cascade or only some of the endogenous CCR7 chemokines are warranted. However, on the other hand, CCR7 agonists or positive allosteric modulators could be warranted in other situations. 

Thus, as a counterpart to the antagonists, it would be favorable to find drugs that could increase the sensitivity of CCR7 towards CCL21, the major LN homing chemokine. Such drugs could perhaps increase DC–T cell priming through increased encounters between these cells in the LN, which could be valuable as a DC vaccine booster to rescue the low efficacy of current cancer vaccines. As G-protein-coupled receptors are excellent drug targets [201] and several drugs already exist on the marked for chemokine receptors (plerixafor, mogamulizumab, and maraviroc) [202], the dual nature of CCR7 targeting with not only one but two drug modalities holds great promise for future CCR7-based therapeutics.

## Figures and Tables

**Figure 1 ijms-22-08340-f001:**
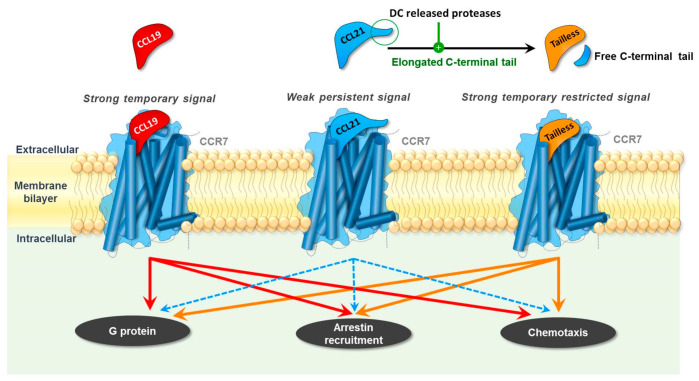
CCL19, CCL21, and Tailless-CCL21 (Tailless) induce differential signaling through their common receptor CCR7. Overall, CCL19 is a strong agonist of both G-protein signaling, β-arrestin recruitment, and chemotaxis, whereas CCL21 is a weak agonist. Upon cleavage by DC-released proteases, CCL21 is turned into Tailless-CCL21, which resembles CCL19 and, thus, is a strong agonist.

**Figure 2 ijms-22-08340-f002:**
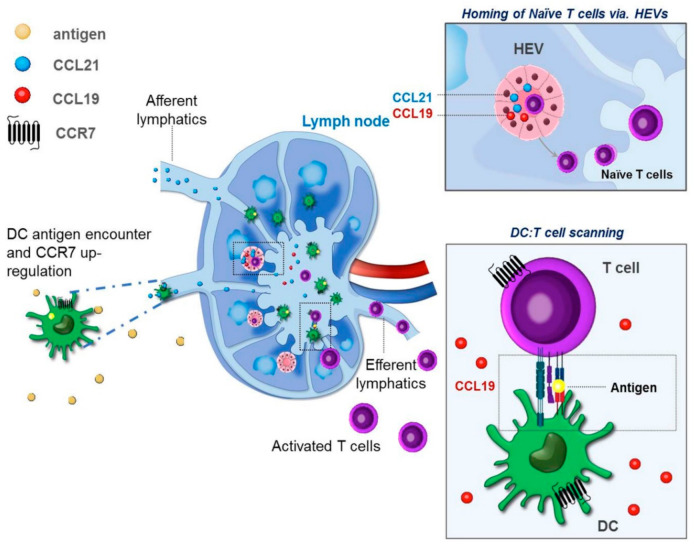
Based on their differential tissue expression and signaling properties, the chemokines CCL19 and CCL21 exert different functions in vivo. CCL21 is believed to be the major lymph node (LN)-homing chemokine, directing the LN localization of activated CCR7^+^ dendritic cells (DCs) through afferent lymphatics. As both CCL19 and CCL21 are present in high endothelial venules (HEVs) and T cells enter the LN through the HEVs, both CCL19 and CCL21 appear to be important for directing the localization of CCR7^+^ T cell subsets (naïve, memory, and Treg) to the LN. CCL19 secreted by active DCs in the LN partakes in the subsequent scanning and DC–T cell priming process, directing T cells to interact with the DCs. The rapid internalization of CCR7 occurring upon receptor engagement with CCL19 allows for a swift DC–T cell interaction and allowing for the scanning process to occur.

**Figure 3 ijms-22-08340-f003:**
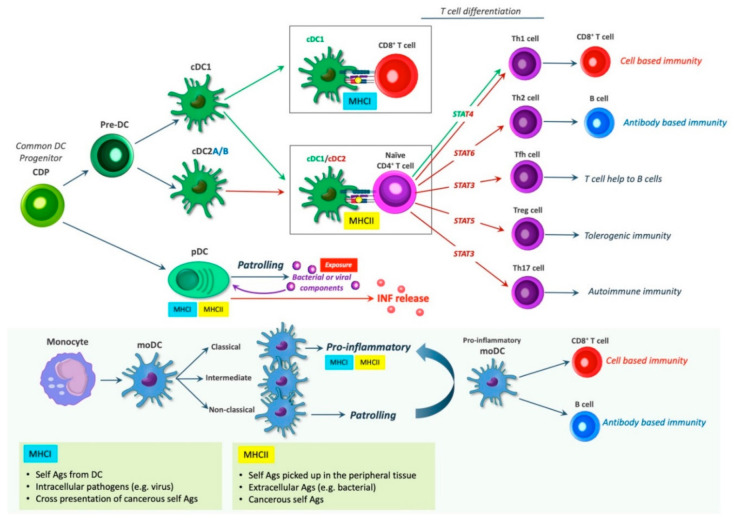
DC subsets and T-cell differentiation. Various DC types exist that partake in regulating immune reactions. Depending on the context, DCs may induce the activation of different T cell subsets. cDC1s can present Ags on MHCII to naïve CD4^+^ T cells to induce Th1 response or cross-present Ags on MHCI to cytotoxic CD8^+^ T cells, both resulting in cell-based immunity. In mice, cDC1s and cDC2s can induce Th1 cell and Th17 cell skewing, respectively, whereas human cDC2s can induce both Th1 cell and Th17 cell skewing [69]. cDC2s induces the T-cell differentiation of naïve CD4^+^ T cells through MHCII Ag presentation, giving rise to the T cell subsets: Th1, Th17, Th2, Tfh, and Treg, which induce cell-based immunity, autoimmune immunity, antibody based immunity, T cell help to B cells, and tolerogenic immunity, respectively. pDCs express both MHCI and MHCII and patrol the bloodstream and peripheral lymphoid organs. When exposed to viral and bacterial infections, pDCs produce type I interferon (IFN). MoDCs are divided into classical, intermediate and non-classical subsets, of which the classical seem to be pro-inflammatory, while the non-classical subset exerts patrolling functions. The pro-inflammatory moDCs induce cell based- and antibody-based immunity via CD8^+^ T cells and B cells, respectively.

**Figure 4 ijms-22-08340-f004:**
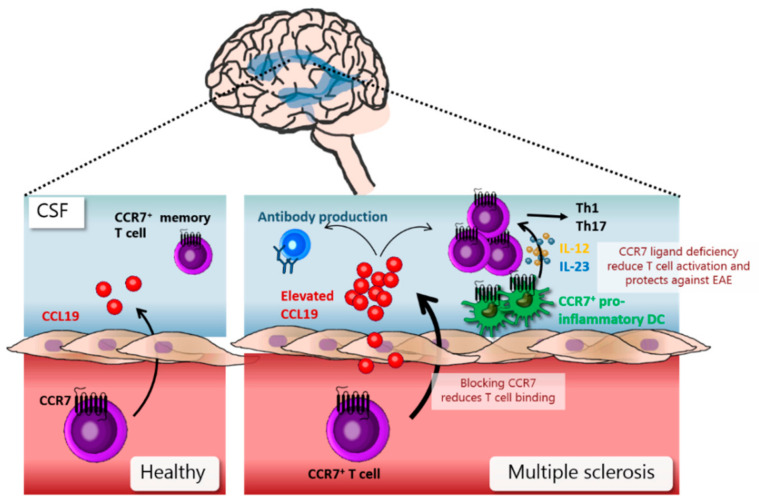
CCR7 and dendritic cells (DCs) in multiple sclerosis (MS): In healthy individuals, CCR7 and CCL19 are involved in the normal trafficking of memory T cells into the brain. In MS, elevated levels of CCL19 can be found in the CSF (cerebrospinal fluid). MS is associated with an increase in T cell infiltrates and an increase in pro-inflammatory CCR7^+^ DCs in the CSF. Blocking CCR7 reduces the binding of T cells to the endothelium and recruitment into the CSF. CCR7 signaling in DCs leads to the release of IL-12 and IL-23, which, in turn, induces T-cell activation. Deficiency of CCR7 ligands reduce T-cell activation and protects against EAE in the EAE mouse model.

**Figure 5 ijms-22-08340-f005:**
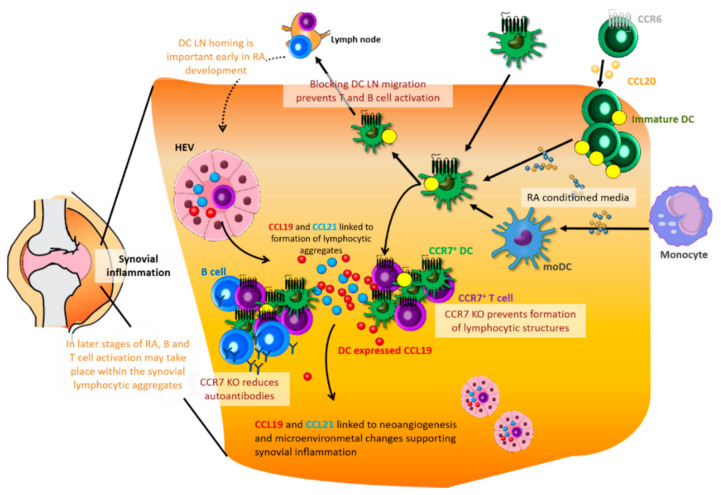
Dendritic cells (DCs) and the CCR7 axis in rheumatoid arthritis: DCs can be recruited into the joint but can also differentiate and mature from precursors or monocytes in response to locally produced cytokines. Blocking of the DC LN migration (by blocking MMP-9) prevents T- and B-cell activation, and displays protection against early RA in a mouse model. DCs from RA lesions can display increased CCL19 production and the presence of CCL21 and CCL19 chemokines are associated with the formation of lymphocytic aggregates both with and without B cells. These aggregates can support T- and B-cell activation in the later stages of RA. CCR7 KO protects against the formation of the lymphocytic aggregates in a mouse model and can also reduce the production of autoantibodies. CCL21 and CCL19 are also associated with microenvironmental changes, which can support the synovial inflammation.

**Figure 6 ijms-22-08340-f006:**
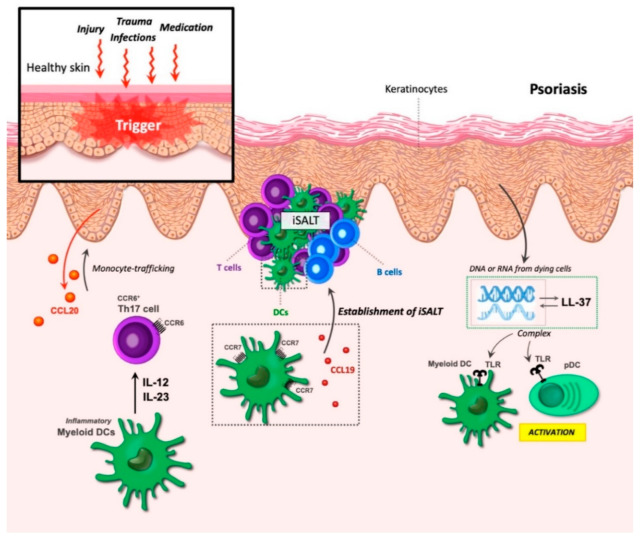
Dendritic cells (DCs) in psoriasis. Psoriasis can be triggered by various factors, including trauma, infections, injury, and medication. The increased proliferation of keratinocytes leads to thickening of the epidermis, while the corneocytes fail to stack normally. During epidermal injury, keratinocytes secrete CCL20, leading to the recruitment of CCR6^+^ monocytes that, together with dermal DCs, are critical for psoriasis development. Inflammatory myeloid DCs produce the cytokines IL-12 and IL-23, resulting in the activation and expansion of Th17 cells and leading to an autoimmune response. The formation of inducible skin-associated lymphoid tissue (iSALT) structures occurs in psoriatic lesions. CCR7 and CCL19 partake in the establishment of these structures. iSALT primary contains CD3^+^ T cells, CD11c^+^ LAMP3/DC-LAMP^+^ DC, and, to a minor extent, CXCR5^+^ B cells. Keratinocytes produce the antimicrobial peptide LL-37, which can form complexes with DNA and RNA from dying cells. These complexes bind to toll-like receptors (TLRs) in myeloid DCs and pDCs leading to the activation of these cells.

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
