# Peer review of "Dendritic Cells and CCR7 Expression: An Important Factor for Autoimmune Diseases, Chronic Inflammation, and Cancer"

_ijms, 2021, doi:10.3390/ijms22158340_

Round 1
Reviewer 1 Report
The authors summarize the chemokine receptor CCR7 and its ligands CCL19 and CCL21, their association with dendritic cells and T cells, and multiple sclerosis, rheumatoid arthritis, and psoriasis as autoimmune diseases. did. It is a very well-written review, and details such as classification of dendritic cells are particularly significant. Assuming that the main significance of this review is the association with CCR7, dendritic cells and T cells in the three diseases listed above, a summarized schema is depicted in the sub-chapter of each disease. Then, I assume that it will be very happy for the reader. Hopefully, each chapter will add a brief pathology and a diagram of the role of CCR7.
Author Response
Thank you very much for your time and your comments.
We agree a summarization of the three diseases in figures would be very suitable for the readers. Therefore, we have added figures to the three diseases; MS, RA, and psoriasis (figure 4, 5 and 6). These summarize the findings from the articles and how CCR7 and its ligands are implicated in the disease conditions.
We hope these revisions meet your expectations.
Reviewer 2 Report
Authors reviewed human dendritic cells (DCs) and the central chemokine, CCR7 expression in autoimmune diseases and chronic inflammation.
This approach is interesting and instructive to know CCL19 and CCL21 inducing differentiation through their common CCR7 receptor, as well as functional relation to T cell interaction at the priming phase of lymph nodes.
The article is too long to follow like a book chapter. The biology of dendritic cell and T cell interaction from sections 3 to 7 would be indirect background of this article, therefore, authors need to revise them to be a compact form more using MIDP format including references edited correctly.
As 7 to 9 sections are the main issues, it is better to add another Figure summarized like Figure 1 and 2. Authors need to provide description of CCR7 and cancer immunology in section 10 rather than general information.
Figure 3 is incorrect explanation and description in the text about cCD1 and cCD2, please cite such references as Immunology 2018 May;154(1):3-20, Science. 2017 Jun 9;356(6342):eaag3009, Science. 2017 Apr 21;356(6335):eaah4573, and Cell. 2019 Oct 31;179(4):846-863.e24, and so on.
The CCR7 marker on the T cell in Figure 1 at the lower right panel needs to be added for recognition.
English sentences are required to be edited, because the quality seems to be different among sections, and a unified description is necessary for the document though the article.
Author Response
Thank you very much for your time and your comments.
We acknowledge that the manuscript was too long. We have revised sections 3 to 7. To shorten the background information for this article, we have shortened these sections and removed the descriptions on DC surface markers and details regarding T cell differentiation (e.g. transcription factors etc.). Instead, we have tried to incorporate the T cell descriptions, regarding DC induction of Th subtypes, in figure 3.
We agree a summarization of the three diseases in figures would be very suitable for the readers. Therefore, we have added figures to the three diseases; MS, RA, and psoriasis (figure 4, 5 and 6). These summarize the findings from the articles and how CCR7 and its ligands are implicated in the disease conditions.
Figure 3 was incorrect regarding cDC1s and cDC2s, as well as the description of these subtypes in the text. We acknowledge these mistakes, as these could appear misleading to the reader, e.g., as it could incorrectly seem as cDC1s presented Ags on MHCI to Th cells to induce Th1 differentiation. We have revised the figure and the text and have corrected these mistakes in accordance with the references you mentioned.
We have added the CCR7 marker on the T cell in figure 2 (at the lower right panel).
In general, we have edited our English sentences, focusing on grammar, spelling mistakes, and quality.
We hope these revisions meet your expectations.
Round 2
Reviewer 2 Report
Authors revised the manuscript fully met as requested for better understanding of dendritic cell and CCR7 expression.
Additional Figures are so fine; therefore, this article would be expected for quotation in other articles.